# MULTI-SCALE REPRESENTATION LEARNING FOR SPATIAL FEATURE DISTRIBUTIONS USING GRID CELLS

**Gengchen Mai[1], Krzysztof Janowicz[1], Bo Yan[2], Rui Zhu[1], Ling Cai[1] & Ni Lao[3]**
[1]STKO Lab, University of California, Santa Barbara, CA, USA, 93106
{gengchen_mai,janowicz,ruizhu,lingcai}@ucsb.edu
[2]LinkedIn Corporation, Mountain View, CA, USA, 94043
boyan1@linkedin.com
[3]SayMosaic Inc., Palo Alto, CA, USA, 94303
ni.lao@mosaix.ai

## ABSTRACT

Unsupervised text encoding models have recently fueled substantial progress in Natural Language Processing (NLP). The key idea is to use neural networks to convert words in texts to vector space representations (embeddings) based on word positions in a sentence and their contexts, which are suitable for end-to-end training of downstream tasks. We see a strikingly similar situation in spatial analysis, which focuses on incorporating both absolute positions and spatial contexts of geographic objects such as Points of Interest (POIs) into models. A general-purpose representation model for space is valuable for a multitude of tasks. However, no such general model exists to date beyond simply applying discretization or feedforward nets to coordinates, and little effort has been put into jointly modeling distributions with vastly different characteristics, which commonly emerges from GIS data. Meanwhile, Nobel Prize-winning Neuroscience research shows that grid cells in mammals provide a multi-scale periodic representation that functions as a metric for location encoding and is critical for recognizing places and for path-integration. Therefore, we propose a representation learning model called Space2Vec to encode the absolute positions and spatial relationships of places. We conduct experiments on two real-world geographic data for two different tasks: 1) predicting types of POIs given their positions and context, 2) image classification leveraging their geo-locations. Results show that because of its multi-scale representations, Space2Vec outperforms well-established ML approaches such as RBF kernels, multi-layer feed-forward nets, and tile embedding approaches for location modeling and image classification tasks. Detailed analysis shows that all baselines can at most well handle distribution at one scale but show poor performances in other scales. In contrast, Space2Vec 's multi-scale representation can handle distributions at different scales. [1]

## 1 INTRODUCTION

Unsupervised text encoding models such as Word2Vec (Mikolov et al., 2013), Glove (Pennington et al., 2014), ELMo (Peters et al., 2018), and BERT (Devlin et al., 2018) have been effectively utilized in many Natural Language Processing (NLP) tasks. At their core they train models which encode words into vector space representations based on their positions in the text and their context. A similar situation can be encountered in the field of Geographic Information Science (GIScience). For example, spatial interpolation aims at predicting an attribute value, e.g., elevation, at an unsampled location based on the known attribute values of nearby samples. Geographic information has become an important component to many tasks such as fine-grained image classification (Mac Aodha et al., 2019), point cloud classification and semantic segmentation (Qi et al., 2017), reasoning about Point of Interest (POI) type similarity (Yan et al., 2017), land cover classification (Kussul et al., 2017), and geographic question answering (Mai et al., 2019b). Developing a *general* model for vector space representation of any point in space would pave the way for many future applications.

---

[1]Link to project repository: https://github.com/gengchenmai/space2vec

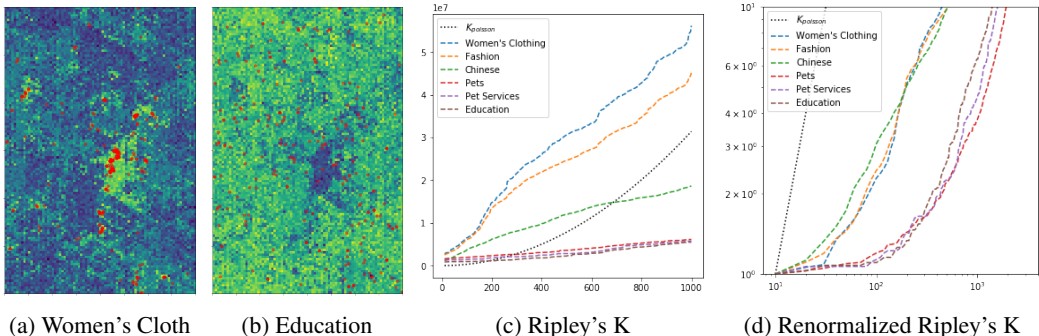

(a) Women's Cloth     (b) Education     (c) Ripley's K     (d) Renormalized Ripley's K

Figure 1: The challenge of joint modeling distributions with very different characteristics. (a)(b) The POI locations (red dots) in Las Vegas and Space2Vec predicted conditional likelihood of Women's Clothing (with a clustered distribution) and Education (with an even distribution). The dark area in (b) indicates that the downtown area has more POIs of other types than education. (c) Ripley's K curves of POI types for which Space2Vec has the largest and smallest improvement over $wrap$ (Mac Aodha et al., 2019). Each curve represents the number of POIs of a certain type inside certain radios centered at every POI of that type; (d) Ripley's K curves renormalized by POI densities and shown in log-scale. To efficiently achieve multi-scale representation Space2Vec concatenates the grid cell encoding of 64 scales (with wave lengths ranging from 50 meters to $40k$ meters) as the first layer of a deep model, and trains with POI data in an unsupervised fashion.

However, existing models often utilize *specific* methods to deal with geographic information and often disregards geographic coordinates. For example, Place2Vec (Yan et al., 2017) converts the coordinates of POIs into spatially collocated POI pairs within certain distance bins, and does not preserve information about the (cardinal) direction between points. Li et al. (2017) propose DCRNN for traffic forecasting in which the traffic sensor network is converted to a distance weighted graph which necessarily forfeits information about the spatial layout of sensors. There is, however, no general representation model beyond simply applying discretization (Berg et al., 2014; Tang et al., 2015) or feed-forward nets (Chu et al., 2019; Mac Aodha et al., 2019) to coordinates.

A key challenge in developing a general-purpose representation model for space is how to deal with mixtures of distributions with very different characteristics (see an example in Figure 1), which often emerges in spatial datasets (McKenzie et al., 2015). For example, there are POI types with clustered distributions such as women's clothing, while there are other POI types with regular distributions such as education. These feature distributions co-exist in the same space, and yet we want a single representation to accommodate all of them in a task such as location-aware image classification (Mac Aodha et al., 2019). Ripley's K is a spatial analysis method used to describe point patterns over a given area of interest. Figure 1c shows the K plot of several POI types in Las Vegas. One can see that as the radius grows the numbers of POIs increase at different rates for different POI types. In order to see the relative change of density at different scales, we renormalize the curves by each POI type's density and show it in log scale in Figure 1d. One can see two distinct POI type groups with different distribution patterns with clustered and even distributions. If we want to model the distribution of these POIs by discretizing the study area into tiles, we have to use small grid sizes for women's clothing while using larger grid sizes for educations because smaller grid sizes lead to over- parameterization of the model and overfitting. In order to jointly describe these distributions and their patterns, we need an encoding method which supports *multi-scale representations*.

Nobel Prize winning Neuroscience research (Abbott & Callaway, 2014) has demonstrated that grid cells in mammals provide a multi-scale periodic representation that functions as a metric for location encoding, which is critical for integrating self-motion. Moreover, Blair et al. (2007) show that the multi-scale periodic representation of grid cells can be simulated by summing three cosine grating functions oriented $60°$ apart, which may be regarded as a simple Fourier model of the hexagonal lattice. This research inspired us to encode locations with multi-scale periodic representations. Our assumption is that decomposed geographic coordinates helps machine learning models, such as deep neural nets, and multi-scale representations deal with the inefficiency of intrinsically single-scale methods such as RFB kernels or discretization (tile embeddings). To validate this intuition, we propose an encoder-decoder framework to encode the distribution of point-features[2] in space and

---

[2]In GIS and spatial analysis, 'features' are representations of real-world entities. A tree can, for instance, be modeled by a point-feature, while a street would be represented as a line string feature.

train such a model in an unsupervised manner. This idea of using sinusoid functions with different frequencies to encode positions is similar to the position encoding proposed in the Transformer model (Vaswani et al., 2017). However, the position encoding model of Transformer deals with a discrete 1D space – the positions of words in a sentence – while our model works on higher dimensional continuous spaces such as the surface of earth.

**In summary, the contributions of our work are as follows:**

1. We propose an encoder-decoder encoding framework called Space2Vec using sinusoid functions with different frequencies to model absolute positions and spatial contexts. We also propose a multi-head attention mechanism based on context points. To the best of our knowledge, this is the first attention model that explicitly considers the spatial relationships between the query point and context points.

2. We conduct experiments on two real world geographic data for two different tasks: 1) predicting types of POIs given their positions and context, 2) image classification leveraging their geo-locations. Space2Vec outperforms well-established encoding methods such as RBF kernels, multi-layer feed-forward nets, and tile embedding approaches for location modeling and image classification.

3. To understand the advantages of Space2Vec we visualize the firing patterns (response maps) of location models' encoding layer neurons and show how they handle spatial structures at different scales by integrating multi-scale representations. Furthermore the firing patterns for the spatial context models neurons give insight into how the grid-like cells capture the decreasing distance effect with multi-scale representations.

## 2 PROBLEM FORMULATION

*Distributed representation of point-features in space* can be formulated as follows. Given a set of points $\mathcal{P} = \{p_i\}$, i.e., Points of Interests (POIs), in $L$-D space ($L = 2, 3$) define a function $f_{\mathcal{P},\theta}(\mathbf{x}) : \mathbb{R}^L \to \mathbb{R}^d$ ($L \ll d$), which is parameterized by $\theta$ and maps any coordinate $\mathbf{x}$ in space to a vector representation of $d$ dimension. Each point (e.g., a restaurant) $p_i = (\mathbf{x}_i, \mathbf{v}_i)$ is associated with a location $\mathbf{x}_i$ and attributes $\mathbf{v}_i$ (i.e., POI features such as type, name, capacity, etc.). The function $f_{\mathcal{P},\theta}(\mathbf{x})$ encodes the probability distribution of point features over space and can give a representation of any point in the space. Attributes (e.g. place types such as *Museum*) and coordinate of point can be seen as analogies to words and word positions in commonly used word embedding models.

## 3 RELATED WORK

There has been theoretical research on neural network based path integration/spatial localization models and their relationships with grid cells. Both Cueva & Wei (2018) and Banino et al. (2018) showed that grid-like spatial response patterns emerge in trained networks for navigation tasks which demonstrate that grid cells are critical for vector-based navigation. Moreover, Gao et al. (2019) propose a representational model for grid cells in navigation tasks which has good quality such as magnified local isometry. All these research is focusing on understanding the relationship between the grid-like spatial response patterns and navigation tasks from a theoretical perspective. In contrast, our goal focuses on utilizing these theoretical results on real world data in geoinformatics.

Radial Basis Function (RBF) kernel is a well-established approach to generating learning friendly representation from points in space for machine learning algorithms such as SVM classification (Baudat & Anouar, 2001) and regression (Bierens, 1994). However, the representation is example based – i.e., the resultant model uses the positions of training examples as the centers of Gaussian kernel functions (Maz'ya & Schmidt, 1996). In comparison, the grid cell based location encoding relies on sine and cosine functions, and the resultant model is inductive and does not store training examples.

Recently the computer vision community shows increasing interests in incorporating geographic information (e.g. coordinate encoding) into neural network architectures for multiple tasks such as image classification (Tang et al., 2015) and fine grained recognition (Berg et al., 2014; Chu et al., 2019; Mac Aodha et al., 2019). Both Berg et al. (2014) and Tang et al. (2015) proposed to discretize the study area into regular grids. To model the geographical prior distribution of the image categories, the grid id is used for GPS encoding instead of the raw coordinates. However, choosing the correct discretization is challenging (Openshaw, 1984; Fotheringham & Wong, 1991),

and incorrect choices can significantly affect the final performance (Moat et al., 2018; Lechner et al., 2012). In addition, discretization does not scale well in terms of memory use. To overcome these difficulties, both Chu et al. (2019) and Mac Aodha et al. (2019) advocated the idea of inductive location encoders which directly encode coordinates into a location embedding. However, both of them directly feed the coordinates into a feed-forward neural network (Chu et al., 2019) or residual blocks (Mac Aodha et al., 2019) without any feature decomposition strategy. Our experiments show that this direct encoding approach is insufficient to capture the spatial feature distribution and Space2Vec significantly outperforms them by integrating spatial representations of different scales.

## 4 METHOD

We solve *distributed representation of point-features in space* (defined in Section 2) with an encoder-decoder architecture:

1. Given a point $p_i = (\mathbf{x}_i, \mathbf{v}_i)$ a **point space encoder** $Enc^{(x)}()$ encodes location $\mathbf{x}_i$ into a location embedding $\mathbf{e}[\mathbf{x}_i] \in \mathbb{R}^{d^{(x)}}$ and a **point feature encoder** $Enc^{(v)}()$ encodes its feature into a feature embedding $\mathbf{e}[\mathbf{v}_i] \in \mathbb{R}^{d^{(v)}}$. $\mathbf{e} = [\mathbf{e}[\mathbf{x}_i]; \mathbf{e}[\mathbf{v}_i]] \in \mathbb{R}^d$ is the full representation of point $p_i \in \mathcal{P}$, where $d = d^{(x)} + d^{(v)}$. $[;]$ represents vector concatenation. In contrast, geographic entities not in $\mathcal{P}$ within the studied space can be represented by their location embedding $\mathbf{e}[\mathbf{x}_j]$ since its $\mathbf{v}_i$ is unknown.

2. We developed two types of decoders which can be used independently or jointly. A **location decoder** $Dec_s()$ reconstructs point feature embedding $\mathbf{e}[\mathbf{v}_i]$ given location embedding $\mathbf{e}[\mathbf{x}_i]$, and a **spatial context decoder** $Dec_c()$ reconstructs the feature embedding $\mathbf{e}[\mathbf{v}_i]$ of point $p_i$ based on the space and feature embeddings $\{\mathbf{e}_{i1}, ..., \mathbf{e}_{ij}, ..., \mathbf{e}_{in}\}$ of nearest neighboring points $\{p_{i1}, ..., p_{ij}, ..., p_{in}\}$, where $n$ is a hyper-parameter.

### 4.1 ENCODER

**Point Feature Encoder**     Each point $p_i = (\mathbf{x}_i, \mathbf{v}_i)$ in a point set $\mathcal{P}$ is often associated with features such as the air pollution station data associate with some air quality measures, a set of POIs with POI types and names, a set of points from survey and mapping with elevation values, a set of points from geological survey with mineral content measure, and so on. The point feature encoder $Enc^{(v)}()$ encodes such features $\mathbf{v}_i$ into a feature embedding $\mathbf{e}[\mathbf{v}_i] \in \mathbb{R}^{d^{(v)}}$. The implementation of $Enc^{(v)}()$ depends on the nature of these features. For example, if each point represents a POI with multiple POI types (as in this study), the feature embedding $\mathbf{e}[\mathbf{v}_i]$ can simply be the mean of each POI types' embeddings $\mathbf{e}[\mathbf{v}_i] = \frac{1}{H} \sum_{h=1}^{H} \mathbf{t}_h^{(\gamma)}$, where $\mathbf{t}_h^{(\gamma)}$ indicates the $h$th POI type embedding of a POI $p_i$ with $H$ POI types. We apply $L_2$ normalization to the POI type embedding matrix.

**Point Space Encoder**     A part of the novelty of this paper is from the point space encoder $Enc^{(x)}()$. We first introduce Theorem 1 which provide an analytical solution $\phi(\mathbf{x})$ as the base of encoding any location $\mathbf{x} \in \mathbb{R}^2$ in 2D space to a distributed representation:

**Theorem 1.** *Let* $\Psi(\mathbf{x}) = (e^{i\langle \mathbf{a}_j, \mathbf{x} \rangle}, j = 1, 2, 3)^T \in \mathbb{C}^3$ *where* $e^{i\theta} = \cos\theta + i\sin\theta$ *is the Euler notation of complex values;* $\langle \mathbf{a}_j, \mathbf{x} \rangle$ *is the inner product of* $\mathbf{a}_j$ *and* $\mathbf{x}$. $\mathbf{a}_1, \mathbf{a}_2, \mathbf{a}_3 \in \mathbb{R}^2$ *are 2D vectors such that the angle between* $\mathbf{a}_k$ *and* $\mathbf{a}_l$ *is* $2\pi/3$, $\forall j$, $\|\mathbf{a}_j\| = 2\sqrt{\alpha}$. *Let* $\mathbf{C} \in \mathbb{C}^{3 \times 3}$ *be a random complex matrix such as* $\mathbf{C}^*\mathbf{C} = \mathbf{I}$. *Then* $\phi(\mathbf{x}) = \mathbf{C}\Psi(\mathbf{x})$, $M(\Delta\mathbf{x}) = \mathbf{C}diag(\Psi(\Delta\mathbf{x}))\mathbf{C}^*$ *satisfies*

$$\phi(\mathbf{x} + \Delta\mathbf{x}) = M(\Delta\mathbf{x})\phi(\mathbf{x}) \tag{1}$$

*and*

$$\langle \phi(\mathbf{x} + \Delta\mathbf{x}), \phi(\mathbf{x}) \rangle = d(1 - \alpha\|\Delta\mathbf{x}\|^2) \tag{2}$$

*where* $d = 3$ *is the dimension of* $\phi(\mathbf{x})$ *and* $\Delta\mathbf{x}$ *is a small displacement from* $\mathbf{x}$.

The proof of Theorem 1 can be seen in Gao et al. (2019). $\phi(\mathbf{x}) = \mathbf{C}\Psi(\mathbf{x}) \in \mathbb{C}^3$ amounts to a 6-dimension real value vector and each dimension shows a *hexagon* firing pattern which models the grid cell behavior. Because of the periodicity of $sin()$ and $cos()$, this single scale representation $\phi(\mathbf{x})$ does not form a global codebook of 2D positions, i.e. there can be $\mathbf{x} \neq \mathbf{y}$, but $\phi(\mathbf{x}) = \phi(\mathbf{y})$.

Inspired by Theorem 1 and the multi-scale periodic representation of grid cells in mammals (Abbott & Callaway, 2014) we set up our point space encoder $\mathbf{e}[\mathbf{x}] = Enc_{theory}^{(x)}(\mathbf{x})$ to use sine

and cosine functions of different frequencies to encode positions in space. Given any point $\mathbf{x}$ in the studied 2D space, the space encoder $Enc^{(x)}_{theory}(\mathbf{x}) = \mathbf{NN}(PE^{(t)}(\mathbf{x}))$ where $PE^{(t)}(\mathbf{x}) = [PE^{(t)}_0(\mathbf{x}); ...; PE^{(t)}_s(\mathbf{x}); ...; PE^{(t)}_{S-1}(\mathbf{x})]$ is a concatenation of multi-scale representations of $d^{(x)} = 6S$ dimensions. Here $S$ is the total number of grid scales and $s = 0, 1, 2, ..., S-1$. $\mathbf{NN}()$ represents fully connected ReLU layers. Let $\mathbf{a}_1 = [1, 0]^T, \mathbf{a}_2 = [-1/2, \sqrt{3}/2]^T, \mathbf{a}_3 = [-1/2, -\sqrt{3}/2]^T \in \mathbb{R}^2$ be three unit vectors and the angle between any of them is $2\pi/3$. $\lambda_{min}, \lambda_{max}$ are the minimum and maximum grid scale and $g = \frac{\lambda_{max}}{\lambda_{min}}$. At each scale $s$, $PE^{(t)}_s(\mathbf{x}) = [PE^{(t)}_{s,1}(\mathbf{x}); PE^{(t)}_{s,2}(\mathbf{x}); PE^{(t)}_{s,3}(\mathbf{x})]$ is a concatenation of three components, where

$$PE^{(t)}_{s,j}(\mathbf{x}) = [\cos(\frac{\langle \mathbf{x}, \mathbf{a}_j \rangle}{\lambda_{min} \cdot g^{s/(S-1)}}); \sin(\frac{\langle \mathbf{x}, \mathbf{a}_j \rangle}{\lambda_{min} \cdot g^{s/(S-1)}})] \forall j = 1, 2, 3; \tag{3}$$

$\mathbf{NN}()$ and $PE^{(t)}(\mathbf{x})$ are analogies of $\mathbf{C}$ and $\mathbf{\Psi}(\mathbf{x})$ in Theorem 1.

Similarly we can define another space encoder $Enc^{(x)}_{grid}(\mathbf{x}) = \mathbf{NN}(PE^{(g)}(\mathbf{x}))$ inspired by the position encoding model of Transformer (Vaswani et al., 2017), where $PE^{(g)}(\mathbf{x}) = [PE^{(g)}_0(\mathbf{x}); ...; PE^{(g)}_s(\mathbf{x}); ...; PE^{(g)}_{S-1}(\mathbf{x})]$ is still a concatenation of its multi-scale representations, while $PE^{(g)}_s(\mathbf{x}) = [PE^{(g)}_{s,1}(\mathbf{x}); PE^{(g)}_{s,2}(\mathbf{x})]$ handles each component $l$ of $\mathbf{x}$ separately:

$$PE^{(g)}_{s,l}(\mathbf{x}) = [\cos(\frac{\mathbf{x}^{[l]}}{\lambda_{min} \cdot g^{s/(S-1)}}); \sin(\frac{\mathbf{x}^{[l]}}{\lambda_{min} \cdot g^{s/(S-1)}})] \forall l = 1, 2 \tag{4}$$

## 4.2 DECODER

Two types of decoders are designed for two major types of GIS problems: location modeling and spatial context modeling (See Section 5.1).

**Location Decoder** $Dec_s()$ directly reconstructs point feature embedding $\mathbf{e}[\mathbf{v}_i]$ given its space embedding $\mathbf{e}[\mathbf{x}_i]$. We use one layer feed-forward neural network $\mathbf{NN}_{dec}()$

$$\mathbf{e}[\mathbf{v}_i]' = Dec_s(\mathbf{x}_i; \theta_{dec_s}) = \mathbf{NN}_{dec}(\mathbf{e}[\mathbf{x}_i]) \tag{5}$$

For training we use inner product to compare the reconstructed feature embedding $\mathbf{e}[\mathbf{v}_i]'$ against the real feature embeddings of $\mathbf{e}[\mathbf{v}_i]$ and other negative points (see training detail in Sec 4.3).

**Spatial Context Decoder** $Dec_c()$ reconstructs the feature embedding $\mathbf{e}[\mathbf{v}_i]$ of the center point $p_i$ based on the space and feature embeddings $\{\mathbf{e}_{i1}, ..., \mathbf{e}_{ij}, ..., \mathbf{e}_{in}\}$ of $n$ nearby points $\{p_{i1}, ..., p_{ij}, ..., p_{in}\}$. Note that the feed-in order of context points should not affect the prediction results, which can be achieved by permutation invariant neural network architectures (Zaheer et al., 2017) like PointNet (Qi et al., 2017).

$$\mathbf{e}[\mathbf{v}_i]' = Dec_c(\mathbf{x}_i, \{\mathbf{e}_{i1}, ..., \mathbf{e}_{ij}, ..., \mathbf{e}_{in}\}; \theta_{dec_c}) = g(\frac{1}{K} \sum_{k=1}^{K} \sum_{j=1}^{n} \alpha_{ijk} \mathbf{e}[\mathbf{v}_{ij}]) \tag{6}$$

Here $g$ is an activation function such as sigmoid. $\alpha_{ijk} = \frac{exp(\sigma_{ijk})}{\sum_{o=1}^{n} exp(\sigma_{iok})}$ is the attention of $p_i$ with its $j$th neighbor through the $k$th attention head, and

$$\sigma_{ijk} = LeakyReLU(\mathbf{a}_k^T[\mathbf{e}[\mathbf{v}_i]_{init}; \mathbf{e}[\mathbf{v}_{ij}]; \mathbf{e}[\mathbf{x}_i - \mathbf{x}_{ij}]]) \tag{7}$$

where $\mathbf{a}_k \in \mathbb{R}^{2d^{(v)} + d^{(x)}}$ is the attention parameter in the $k$th attention head. The multi-head attention mechanism is inspired by Graph Attention Network (Veličković et al., 2018) and Mai et al. (2019a).

To represent the spatial relationship (distance and direction) between each context point $p_{ij} = (\mathbf{x}_{ij}, \mathbf{v}_{ij})$ and the center point $p_i = (\mathbf{x}_i, \mathbf{v}_i)$, we use the space encoder $Enc^{(x)}()$ to encode the displacement between them $\Delta \mathbf{x}_{ij} = \mathbf{x}_i - \mathbf{x}_{ij}$. Note that we are modeling the spatial interactions between the center point and $n$ context points simultaneously.

In Eq. 7, $\mathbf{e}[\mathbf{v}_i]_{init}$ indicates the initial guess of the feature embedding $\mathbf{e}[\mathbf{v}_i]$ of point $p_i$ which is computed by using another multi-head attention layer as Eq. 6 where the weight $\alpha'_{ijk} = \frac{exp(\sigma'_{ijk})}{\sum_{o=1}^{n} exp(\sigma'_{iok})}$. Here, $\sigma'_{ijk}$ is computed as Eq. 8 where the query embedding $\mathbf{e}[\mathbf{v}_i]$ is excluded.

$$\sigma'_{ijk} = LeakyReLU(\mathbf{a}_k'^T[\mathbf{e}[\mathbf{v}_{ij}]; \mathbf{e}[\mathbf{x}_i - \mathbf{x}_{ij}]]) \tag{8}$$

## 4.3 Unsupervised Training

The unsupervised learning task can simply be maximizing the log likelihood of observing the true point $p_i$ at position $\mathbf{x}_i$ among all the points in $\mathcal{P}$

$$\mathcal{L}_{\mathcal{P}}(\theta) = -\sum_{p_i \in \mathcal{P}} \log P(p_i|p_{i1}, ..., p_{ij}, ..., p_{in}) = -\sum_{p_i \in \mathcal{P}} \log \frac{\exp(\mathbf{e}[\mathbf{v}_i]^T \mathbf{e}[\mathbf{v}_i]')}{\sum_{p_o \in \mathcal{P}} \exp(\mathbf{e}[\mathbf{v}_o]^T \mathbf{e}[\mathbf{v}_i]')} \quad (9)$$

Here only the feature embedding of $p_i$ is used (without location embedding) to prevent revealing the identities of the point candidates, and $\theta = [\theta_{\text{enc}}; \theta_{\text{dec}}]$

Negative sampling by Mikolov et al. (2013) can be used to improve the efficiency of training

$$\mathcal{L}'_{\mathcal{P}}(\theta) = -\sum_{p_i \in \mathcal{P}} \left( \log \sigma(\mathbf{e}[\mathbf{v}_i]^T \mathbf{e}[\mathbf{v}_i]') + \frac{1}{|\mathcal{N}_i|} \sum_{p_o \in \mathcal{N}_i} \log \sigma(-\mathbf{e}[\mathbf{v}_o]^T \mathbf{e}[\mathbf{v}_i]') \right) \quad (10)$$

Here $\mathcal{N}_i \subseteq \mathcal{P}$ is a set of sampled negative points for $p_i$ ($p_i \notin \mathcal{N}_i$) and $\sigma(x) = 1/(1 + e^{-x})$.

## 5 Experiment

In this section we compare Space2Vec with commonly used position encoding methods, and analyze them both quantitatively and qualitatively.

**Baselines** Our baselines include 1) *direct* directly applying feed-forward nets (Chu et al., 2019); 2) *tile* discretization (Berg et al., 2014; Adams et al., 2015; Tang et al., 2015); 3) *wrap* feed-forward nets with coordinate wrapping (Mac Aodha et al., 2019); and 4) *rbf* Radial Basis Function (RBF) kernels (Baudat & Anouar, 2001; Bierens, 1994). See Appendix A.1 for details of the baselines.

### 5.1 POI Type Classification Tasks

**Dataset and Tasks** To test the proposed model, we conduct experiments on geographic datasets with POI position and type information. We utilize the open-source dataset published by Yelp Data Challenge and select all POIs within the Las Vegas downtown area[3]. There are 21,830 POIs with 1,191 different POI types in this dataset. Note that each POI may be associated with one or more types, and we do not use any other meta-data such as business names, reviews for this study. We project geographic coordinates into projection coordinates using the NAD83/Conus Albers projection coordinate system[4]. The POIs are split into training, validation, and test dataset with ratios 80%:10%:10%. We create two tasks setups which represent different types of modeling need in Geographic Information Science:

- **Location Modeling** predicts the feature information associated with a POI based on its location $\mathbf{x}_i$ represented by the *location decoder* $Dec_s()$. This represents a large number of location prediction problems such as image fine grained recognition with geographic prior (Chu et al., 2019), and species potential distribution prediction (Zuo et al., 2008).
- **Spatial Context Modeling** predicts the feature information associated with a POI based on its context $\{\mathbf{e}_{i1}, ..., \mathbf{e}_{ij}, ..., \mathbf{e}_{in}\}$ represented by the *spatial context decoder* $Dec_c()$. This represents a collections of spatial context prediction problem such as spatial context based facade image classification (Yan et al., 2018), and all spatial interpolation problems.

We use POI prediction metrics to evaluate these models. Given the real point feature embedding $\mathbf{e}[\mathbf{v}_i]$ and $N$ negative feature embeddings $\mathcal{N}_i = \{\mathbf{e}[\mathbf{v}_i]^-\}$, we compare the predicted $\mathbf{e}[\mathbf{v}_i]'$ with them by cosine distance. The cosine scores are used to rank $\mathbf{e}[\mathbf{v}_i]$ and $N$ negative samples. The negative feature embeddings are the feature embeddings of points $p_j$ randomly sampled from $\mathcal{P}$ and $p_i \neq p_j$. We evaluate each model using Negative Log-Likelihood (NLL), Mean Reciprocal Rank (MRR) and HIT@5 (the chance of the true POI being ranked to top 5. We train and test each model 10 times to estimate standard deviations. See Appendix A.2 for hyper-parameter selection details.

### 5.1.1 Location Modeling Evaluation

We first study location modeling with the **location decoder** $Dec_s()$ in Section 4.2. We use a negative sample size of $N = 100$. Table 1 shows the average metrics of different models with their best hyper-

---

[3]The geographic range is (35.989438, 36.270897) for latitude and (-115.047977, -115.3290609) for longitude.
[4]`https://epsg.io/5070-1252`

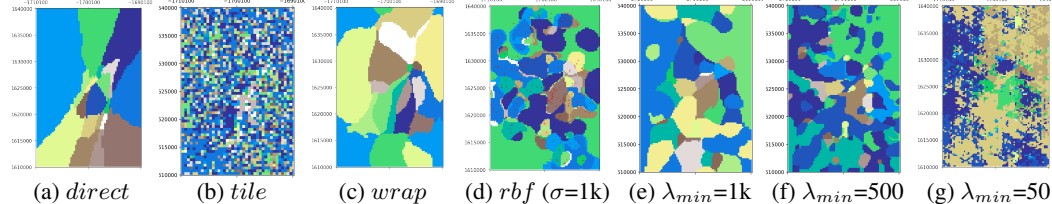

(a) *direct*  (b) *tile*  (c) *wrap*  (d) *rbf* ($\sigma$=1k)  (e) $\lambda_{min}$=1k  (f) $\lambda_{min}$=500  (g) $\lambda_{min}$=50

Figure 2: Embedding clustering of (a) *direct*; (b) *tile* with the best cell size $c = 500$; (c) *wrap* ($h = 3, o = 512$); (d) *rbf* with the best $\sigma$ (1k) and 200 anchor points (red) and (e)(f)(h) *theory* models with different $\lambda_{min}$, but fixed $\lambda_{max} = 40k$ and $S = 64$. All models use 1 hidden ReLU layers of 512 neurons except *wrap*.

Table 1: The evaluation results of different location models on the validation and test dataset.

| | Train NLL | NLL | Validation MRR | HIT@5 | Testing MRR | HIT@5 |
|---|---|---|---|---|---|---|
| *random* | | - | 0.052 (0.002) | 4.8 (0.5) | 0.051 (0.002) | 5.0 (0.5) |
| *direct* | 1.285 | 1.332 | 0.089 (0.001) | 10.6 (0.2) | 0.090 (0.001) | 11.3 (0.2) |
| *tile* (c=500) | 1.118 | 1.261 | 0.123 (0.001) | 16.8 (0.2) | 0.120 (0.001) | 17.1 (0.3) |
| *wrap*(h=3,o=512) | 1.222 | 1.288 | 0.112 (0.001) | 14.6 (0.1) | 0.119 (0.001) | 15.8 (0.2) |
| *rbf* ($\sigma$=1k) | 1.209 | 1.279 | 0.115 (0.001) | 15.2 (0.2) | 0.123 (0.001) | 16.8 (0.3) |
| *grid* ($\lambda_{min}$=50) | 1.156 | 1.258 | 0.128 (0.001) | 18.1 (0.3) | 0.139 (0.001) | **20.0** (0.2) |
| *hexa* ($\lambda_{min}$=50) | 1.230 | 1.297 | 0.107 (0.001) | 14.0 (0.2) | 0.105 (0.001) | 14.5 (0.2) |
| *theorydiag* ($\lambda_{min}$=50) | 1.277 | 1.324 | 0.094 (0.001) | 12.3 (0.3) | 0.094 (0.002) | 11.2 (0.3) |
| *theory* ($\lambda_{min}$=1k) | 1.207 | 1.281 | 0.123 (0.002) | 16.3 (0.5) | 0.121 (0.001) | 16.2 (0.1) |
| *theory* ($\lambda_{min}$=500) | 1.188 | 1.269 | 0.132 (0.001) | 17.6 (0.3) | 0.129 (0.001) | 17.7 (0.2) |
| *theory* ($\lambda_{min}$=50) | 1.098 | 1.249 | **0.137** (0.002) | **19.4** (0.1) | **0.144** (0.001) | **20.0** (0.2) |

parameter setting on the validation set. We can see that *direct* and *theorydiag* are less competitive, only beating the *random* selection baseline. Other methods with single scale representations – including *tile*, *wrap*, and *rbf* – perform better. The best results come from various version of the grid cell models, which are capable of dealing with multi-scale representations.

In order to understand the reason for the superiority of grid cell models we provide qualitative analysis of their representations. We apply hierarchical clustering to the location embeddings produced by studied models using cosine distance as the distance metric (See Fig. 2). we can see that when restricted to large grid sizes ($\lambda_{min} = 1k$), *theory* has similar representation (Fig. 2d, 2e, and Fig. 4d, 4e) and performance compared to *rbf* ($\sigma = 1k$). However it is able to significantly outperform *rbf* ($\sigma = 1k$) (and *tile* and *wrap*) when small grid sizes ($\lambda_{min} = 500, 50$) are available. The relative improvements over *rbf* ($\sigma = 1k$) are -0.2%, +0.6%, +2.1% MRR for $\lambda_{min}$=1k, 500, 50 respectively.

### 5.1.2 MULTI-SCALE ANALYSIS OF LOCATION MODELING

In order to show how our multi-scale location representation model will affect the prediction of POI types with different distribution patterns, we classify all 1,191 POI types into three groups based on radius $r$, which is derived from each POI types' renormalized Ripley's K curve (See Figure 1d for examples). It indicates the x axis value of the intersection between the curve and the line of $y = 3.0$. A lower $r$ indicates a more clustered distribution patterns. These three groups are listed below:

1. Clustered ($r \leqslant 100m$): POI types with clustered distribution patterns;
2. Middle ($100m < r < 200m$): POI types with less extreme scales;
3. Even ($r \geqslant 200m$): POI types with even distribution patterns.

Table 2 shows the performance ($MRR$) of *direct*, *tile*, *wrap*, *rbf*, and our *theory* model on the test dataset of the location modeling task with respect to these three different POI distribution groups. The numbers in () indicate the MRR difference betweeb a baseline and *theory*. *# POI* refers to total number of POI belong to each group[5]. We can see that 1) The two neural net approaches (*direct* and *wrap*) have no scale related parameter and are not performing ideally across all scales, with *direct*

---

[5]The reason why the sum of *# POI* of these three groups does not equal to the total number of POI is because one POI can have multiple types and they may belonging to different groups.

Table 2: Comparing performances in different POI groups. We classify all 1,191 POI types into three groups based on the radius $r$ of their root types, where their renormalized Ripley's K curve (See Figure 1d) reach 3.0: 1) Clustered ($r \leqslant 100m$): POI types with clustered distribution patterns; 2) Middle ($100m < r < 200m$): POI types with unclear distribution patterns; 3) Even ($r \geqslant 200m$): POI types with even distribution patterns. The MRR of $wrap$ and $theory$ on those three groups are shown. The numbers in () indicate the difference between the MRR of a baseline model and the MRR of $theory$ with respect to a specific group. *#POI* refers to the total number of POIs belonging to each group. *Root Types* indicates the root categories of those POI types belong to each group.

| POI Groups | Clustered ($r \leqslant 100m$) | Middle ($100m < r < 200m$) | Even ($r \geqslant 200m$) |
|---|---|---|---|
| $direct$ | 0.080 (-0.047) | 0.108 (-0.030) | 0.084 (-0.047) |
| $wrap$ | 0.106 (-0.021) | 0.126 (-0.012) | 0.122 (-0.009) |
| $tile$ | 0.108 (-0.019) | 0.135 (-0.003) | 0.111 (-0.020) |
| $rbf$ | 0.112 (-0.015) | 0.136 (-0.002) | 0.119 (-0.012) |
| $theory$ | 0.127 (-) | 0.138 (-) | 0.131 (-) |
| # POI | 16,016 | 7,443 | 3,915 |
| Root Types | Restaurants; Shopping; Food; Nightlife; Automotive; Active Life; Arts & Entertainment; Financial Services | Beauty & Spas; Health & Medical; Local Services; Hotels & Travel; Professional Services; Public Services & Government | Home Services; Event Planning & Services; Pets; Education |

performs worse because of its simple single layer network. 2) The two approaches with built-in scale parameter ($tile$ and $rbf$) have to trade off the performance of different scales. Their best parameter settings lead to close performances to that of Space2Vec at the middle scale, while performing poorly in both clustered and regular groups. These observation clearly shows that **all baselines can at most well handle distribution at one scale but show poor performances in other scales. In contrast, Space2Vec's multi-scale representation can handle distributions at different scales.**

### 5.1.3 SPATIAL CONTEXT MODELING EVALUATION

Next, we evaluate the **spatial context decoder** $Dec_c()$ in Sec. 4.2. We use the same evaluation set up as location modeling. The context points are obtained by querying the $n$-th nearest points using PostGIS ($n = 10$). As for validation and test datasets, we make sure the center points are all unknown during the training phase. Table 3 shows the evaluation results of different models for spatial context modeling. The baseline approaches ($direct$, $tile$, $wrap$, $rbf$) generally perform poorly in context modeling. We designed specialized version of these approaches ($polar$, $polar\_tile$, $scaled\_rbf$) with polar coordinates, which lead to significantly improvements. Note that these are models proposed by us specialized for context modeling and therefore are less general than the grid cell approaches.

Table 3: The evaluation results of different spatial context models on the validation and test dataset. All encoders contains a 1 hidden layer FFN. All grid cell encoders set $\lambda_{min}$=10, $\lambda_{max}$=10k.

| $Space2Vec$ | Train NLL | Validation NLL | MRR | HIT@5 | Testing MRR | HIT@5 |
|---|---|---|---|---|---|---|
| $none$ | 1.163 | 1.297 | 0.159 (0.002) | 22.4 (0.5) | 0.167 (0.006) | 23.4 (0.7) |
| $direct$ | 1.151 | 1.282 | 0.170 (0.002) | 24.6 (0.4) | 0.175 (0.003) | 24.7 (0.5) |
| $polar$ | 1.157 | 1.283 | 0.176 (0.004) | 25.4 (0.4) | 0.178 (0.006) | 24.9 (0.1) |
| $tile$ ($c = 50$) | 1.163 | 1.298 | 0.173 (0.004) | 24.0 (0.6) | 0.173 (0.001) | 23.4 (0.1) |
| $polar\_tile(S = 64)$ | 1.161 | 1.282 | 0.173 (0.003) | 25.0 (0.1) | 0.177 (0.001) | 24.5 (0.3) |
| $wrap$ ($h$=2,$o$=512) | 1.167 | 1.291 | 0.159 (0.001) | 23.0 (0.1) | 0.170 (0.001) | 23.9 (0.2) |
| $rbf$ ($\sigma = 50$) | 1.160 | 1.281 | **0.179** (0.002) | 25.2 (0.6) | 0.172 (0.001) | 25.0 (0.1) |
| $scaled\_rbf$ ($\sigma$=40,$\beta$=0.1) | 1.150 | 1.272 | 0.177 (0.002) | **25.7** (0.1) | 0.181 (0.001) | 25.3 (0.1) |
| $grid(\lambda_{min}$=10) | 1.172 | 1.285 | 0.178 (0.004) | 24.9 (0.5) | 0.181 (0.001) | 25.1 (0.3) |
| $hexa$ ($\lambda_{min}$=10) | 1.156 | 1.289 | 0.173 (0.002) | 24.0 (0.2) | 0.183 (0.002) | 25.3 (0.2) |
| $theorydiag$ ($\lambda_{min} = 10$) | 1.156 | 1.287 | 0.168 (0.001) | 24.1 (0.4) | 0.174 (0.005) | 24.9 (0.1) |
| $theory(\lambda_{min}$=200) | 1.168 | 1.295 | 0.159 (0.001) | 23.1 (0.2) | 0.170 (0.001) | 23.2 (0.2) |
| $theory(\lambda_{min}$=50) | 1.157 | 1.275 | 0.171 (0.001) | 24.2 (0.3) | 0.173 (0.001) | 24.8 (0.4) |
| $theory(\lambda_{min}$=10) | 1.158 | 1.280 | 0.177 (0.003) | 25.2 (0.3) | **0.185** (0.002) | **25.7** (0.3) |

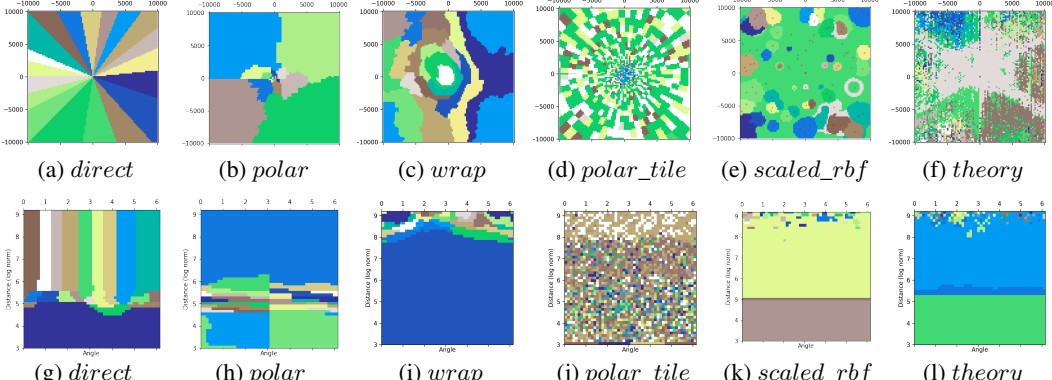

Figure 3: Embedding clustering in the original space of (a) $direct$; (b) $polar$; (c) $wrap$, h=2,o=512; (d) $polar\_tile$, $S = 64$, (e) $scaled\_rbf$, $\sigma = 40$, $\beta$=0.1; and (f) $theory$, $\lambda_{min} = 10$, $\lambda_{max} = 10k$, $S = 64$. (g)(h)(i)(j)(k)(l) are the clustering results of the same models in the polar-distance space using $\log(\parallel \Delta\mathbf{x}_{ij} \parallel +1)$. All models use 1 hidden ReLU (except $wrap$) layers of 512 neurons. Most models except $wrap$ can capture a shift when distance is around $e^5 - 1 \approx 150$ meters.

Nevertheless the grid cell approaches are able to perform better than the specialized approaches on the test dataset while have competitive performance on validation dataset. See Appendix **??** for the visualization of context models. Actually the gains are small for all baseline approaches also. The reason is that we expect location encoding to be less important when context information is accessible. Similarly as discussed in (Gao et al., 2019), it is when there is a lack of visual clues that the grid cells of animals are the most helpful for their navigation.

Figure 6 shows the location embedding clustering results in both Cartesian and polar coordinate systems. We can see that $direct$ (Fig. 3a, 3g) only captures the distance information when the context POI is very close ($log(\parallel \Delta\mathbf{x}_{ij} \parallel +1) \leqslant 5$) while in the farther spatial context it purely models the direction information. $polar$ (Fig. 3b, 3h) has the similar behaviors but captures the distance information in a more fine-grained manner. $wrap$ (Fig. 3c, 3i) mainly focuses on differentiating relative positions in farther spatial context $cont$ which might explain its lower performance[6]. $polar\_tile$ (Fig. 3d) mostly responds to distance information. Interestingly, $scaled\_rbf$ and $theory$ have similar representations in the polar coordinate system (Fig. 3k, 3l) and similar performance (Table 3). While $scaled\_rbf$ captures the gradually decreased distance effect with a scaled kernel size which becomes larger in farther distance, $theory$ achieves this by integrating representations of different scales.

## 5.2 FINE-GRAINED IMAGE CLASSIFICATION TASKS

To demonstrate the generalizability of Space2Vec for space representation we utilized the proposed point space encoder $Enc^{(x)}()$ model in a well-known computer vision task: *fine-grained image classification*. As we discussed in Section 3, many studies (Berg et al., 2014; Chu et al., 2019; Mac Aodha et al., 2019) have shown that geographic prior information - where (and when) the image is taken - is very important additional information for the fine-grained image classification task and can substantially improve the model performance. For example, the appearance information is usually not sufficient to differentiate two visually similar species. In this case, the geographic prior becomes much more important because these two species may have very different spatial prior distributions such as the example of European Toads and Spiny Toads in Figure 1 of Mac Aodha et al. (2019).

We adopt the task setup of Mac Aodha et al. (2019). During training we have a set of tuples $D = \{(I_i, \mathbf{x}_i, y_i, p_i) \mid i = 1, ..., N\}$ where $I_i$ indicates an image, $y_i \in \{1, 2, ..., C\}$ is the corresponding class label (species category), $\mathbf{x}_i = [longitude_i, latitude_i]$ is the geographic coordinates where the image was taken, and $p_i$ is the id of the photographer who took this image. At training time, a location encoder is trained to capture the spatial prior information $P(y \mid \mathbf{x})$. At inference time, $p_i$ information is not available and the final image classification prediction is calculated based on the

---

[6]Note that $wrap$ is original proposed by Mac Aodha et al. (2019) for location modelling, not spatial context modelling. This results indicates $wrap$ is not good at this task.

Table 4: Fine-grained image classification results on two datasets: BirdSnap† and NABirds†. The classification accuracy is calculated by combining image classification predictions $P(y \mid I)$ with different spatial priors $P(y \mid \mathbf{x})$. The $grid$ and $theory$ model use 1 hidden ReLU layers of 512 neurons. The evaluation results of the baseline models are from Table 1 of Mac Aodha et al. (2019).

| | BirdSnap† | NABirds† |
|---|---|---|
| No Prior (i.e. uniform) | 70.07 | 76.08 |
| Nearest Neighbor (num) | 77.76 | 79.99 |
| Nearest Neighbor (spatial) | 77.98 | 80.79 |
| Adaptive Kernel (Berg et al., 2014) | 78.65 | 81.11 |
| $tile$ (Tang et al., 2015) (location only) | 77.19 | 79.58 |
| $wrap$ (Mac Aodha et al., 2019) (location only) | 78.65 | 81.15 |
| $rbf$ ($\sigma$=1k) | 78.56 | 81.13 |
| $grid$ ($\lambda_{min}$=0.0001, $\lambda_{max}$=360, $S = 64$) | **79.44** | 81.28 |
| $theory$ ($\lambda_{min}$=0.0001, $\lambda_{max}$=360, $S = 64$) | 79.35 | **81.59** |

combination of two models: 1) the trained location encoder which captures the spatial priors $P(y \mid \mathbf{x})$ and 2) the pretrained image classification model, InceptionV3 network (Szegedy et al., 2016), which captures $P(y \mid I)$. Bayesian theory has been used to derive the joint distribution $P(y \mid I, \mathbf{x})$. See Mac Aodha et al. (2019) for detail explanation as well as the loss function. Note that while Space2Vec outperforms specialized density estimation methods such as Adaptive Kernel (Berg et al., 2014), it would be interesting to explore early fusion Space2Vec 's representations with the image module.

We use two versions of our point space encoder $Enc^{(x)}()$ model ($grid$, $theory$) as the location encoder to capture the spatial prior information $P(y \mid \mathbf{x})$. The evaluation results of our models as well as multiple baselines are shown in Table 4. We can see that **both $grid$, $theory$ outperform previous models as well as that of Mac Aodha et al. (2019) on two fine-grained image classification datasets with significant sizes: BirdSnap†, NABirds†**. $theory$ shows superiority over $grid$ on NABirds† while fail to outperform $grid$ on BirdSnap†. Note that we only pick baseline models which capture spatial-only prior and drop models which additionally consider time information. Both $grid$ and $theory$ use 1 hidden ReLU layers of 512 neurons for $\mathbf{NN}()$ and they have the same hyperparameters: $\lambda_{min}$=0.0001, $\lambda_{max}$=360, $S = 64$. Like Mac Aodha et al. (2019), the location embedding size $d^{(x)}$ is 1024 and we train the location encoder for 30 epochs. Our implementation is based on the original code[7] of Mac Aodha et al. (2019) for both model training and evaluation phase.

## 6 CONCLUSION

We introduced an encoder-decoder framework as a general-purpose representation model for space inspired by biological grid cells' multi-scale periodic representations. The model is an inductive learning model and can be trained in an unsupervised manner. We conduct two experiments on POI type prediction based on 1) POI locations and 2) nearby POIs. The evaluation results demonstrate the effectiveness of our model. Our analysis reveals that it is the ability to integrate representations of different scales that makes the grid cell models outperform other baselines on these two tasks. In the future, we hope to incorporate the presented framework to more complex GIS tasks such as social network analysis, and sea surface temperature prediction.

ACKNOWLEDGMENTS

The presented work is partially funded by the NSF award 1936677 C-Accel Pilot - Track A1 (Open Knowledge Network): *Spatially-Explicit Models, Methods, And Services For Open Knowledge Networks*, Esri Inc., and Microsoft AI for Earth Grant: *Deep Species Spatio-temporal Distribution Modeling for Biodiversity Hotspot Prediction*. We thank Dr. Ruiqi Gao for discussions about grid cells, Dr. Wenyun Zuo for discussion about species potential distribution prediction and Dr. Yingjie Hu for his suggestions about the introduction section.

---

[7]https://github.com/macaodha/geo_prior/

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

# A APPENDIX

## A.1 BASELINES

To help understand the mechanism of distributed space representation we compare multiple ways of encoding spatial information. Different models use different point space encoder $Enc^{(x)}()$ to encode either location $\mathbf{x}_i$ (for location modeling $loc$) or the displacement between the center point and one context point $\Delta\mathbf{x}_{ij} = \mathbf{x}_i - \mathbf{x}_{ij}$ (for spatial context modeling $cont$)[8].

- $random$ shuffles the order of the correct POI and $N$ negative samples randomly as the predicted ranking. This shows the lower bound of each metrics.

- $direct$ directly encode location $\mathbf{x}_i$ (or $\Delta\mathbf{x}_{ij}$ for $cont$) into a location embedding $\mathbf{e}[\mathbf{x}_i]$ (or $\mathbf{e}[\Delta\mathbf{x}_{ij}]$) using a feed-forward neural networks (FFNs)[9], denoted as $Enc^{(x)}_{direct}(\mathbf{x})$ without decomposing coordinates into a multi-scale periodic representation. This is essentially the GPS encoding method used by Chu et al. (2019). Note that Chu et al. (2019) is not open sourced and we end up implementing the model architecture ourselves.

- $tile$ divides the study area $A_{loc}$ (for $loc$) or the range of spatial context defined by $\lambda_{max}$, $A_{cont}$, (for $cont$) into grids with equal grid sizes $c$. Each grid has an embedding to be used as the encoding for every location $\mathbf{x}_i$ or displacement $\Delta\mathbf{x}_{ij}$ fall into this grid. This is a common practice by many previous work when dealing with coordinate data (Berg et al., 2014; Adams et al., 2015; Tang et al., 2015).

- $wrap$ is a location encoder model recently introduced by Mac Aodha et al. (2019). It first normalizes $\mathbf{x}$ (or $\Delta\mathbf{x}$) into the range $[-1, 1]$ and uses a coordinate wrap mechanism $[\sin(\pi\mathbf{x}^{[l]}); \cos(\pi\mathbf{x}^{[l]})]$ to convert each dimension of $\mathbf{x}$ into 2 numbers. This is then passed through an initial fully connected layer, followed by a series of $h$ residual blocks, each consisting of two fully connected layers ($o$ hidden neurons) with a dropout layer in between. We adopt the official code of Mac Aodha et al. (2019)[10] for this implementation.

- $rbf$ randomly samples $M$ points from the training dataset as RBF anchor points $\{\mathbf{x}^{anchor}_m, m = 1...M\}$ (or samples $M$ $\Delta\mathbf{x}^{anchor}_m$ from $A_{cont}$ for $cont$)[11], and use gaussian kernels $\exp\big(-\dfrac{\|\mathbf{x}_i - \mathbf{x}^{anchor}_m\|^2}{2\sigma^2}\big)$ (or $\exp\big(-\dfrac{\|\Delta\mathbf{x}_{ij} - \Delta\mathbf{x}^{anchor}_m\|^2}{2\sigma^2}\big)$ for $cont$) on each anchor points, where $\sigma$ is the kernel size. Each point $p_i$ has a $M$-dimension RBF feature vector which is fed into a FNN to obtain the spatial embedding. This is a strong baseline for representing floating number features in machine learning models.

- $grid$ as described in Section 4.1 inspired by the position encoding in Transformer (Vaswani et al., 2017).

- $hexa$ Same as $grid$ but use $sin(\theta)$, $sin(\theta + 2\pi/3)$, and $sin(\theta + 4\pi/3)$ in $PE^{(g)}_{s,l}(\mathbf{x})$.

- $theory$ as described in Section 4.1, uses the theoretical models (Gao et al., 2019) as the first layer of $Enc^{(x)}_{theory}(\mathbf{x})$ or $Enc^{(x)}_{theory}(\Delta\mathbf{x}_{ij})$.

- $theorydiag$ further constrains $\mathbf{NN}()$ as a block diagonal matrix, with each scale as a block.

We also have the following baselines which are specific to the spatial context modeling task.

- $none$ the decoder $Dec_c()$ does not consider the spatial relationship between the center point and context points but only the co-locate patterns such as Place2Vec (Yan et al., 2017). That means we drop the $\mathbf{e}[\Delta\mathbf{x}_{ij}]$ from the attention mechanism in Equ. 7 and 8.

- $polar$ first converts the displacement $\Delta\mathbf{x}_{ij}$ into polar coordinates $(r, \theta)$ centered at the center point where $r = log(\|\Delta\mathbf{x}_{ij}\| +1)$. Then it uses $[r, \theta]$ as the input for a FFN to obtain the spatial relationship embedding in Equ. 7. We find out that it has a significant performance improvement over the variation with $r = \|\Delta\mathbf{x}_{ij}\|$.

---

[8]We will use meter as the unit of $\lambda_{min}, \lambda_{max}, \sigma, c$.

[9]we first normalizes $\mathbf{x}$ (or $\Delta\mathbf{x}$) into the range $[-1, 1]$

[10]http://www.vision.caltech.edu/~macaodha/projects/geopriors/

[11]these anchor points are fixed in both $loc$ and $cont$.

- $polar\_tile$ is a modified version of $tile$ but the grids are extracted from polar coordinates $(r, \theta)$ centered at the center point where $r = log(\| \Delta\mathbf{x}_{ij} \| + 1)$. Instead of using grid size $c$, we use the number of grids along $\theta$ (or $r$) axis, $F$, as the only hyperparameter. Similarly, We find that $r = log(\| \Delta\mathbf{x}_{ij} \| + 1)$ outperform $r = \| \Delta\mathbf{x}_{ij} \|$ significantly.

- $scaled\_rbf$ is a modified version of $rbf$ for $cont$ whose kernel size is proportional to the distance between the current anchor point and the origin, $\| \Delta\mathbf{x}_m^{anchor} \|$. That is $exp\left( -\dfrac{\| \Delta\mathbf{x}_{ij} - \Delta\mathbf{x}_m^{anchor} \|^2}{2\sigma_{scaled}^2}\right)$. Here $\sigma_{scaled} = \sigma + \beta \| \Delta\mathbf{x}_m^{anchor} \|$ where $\sigma$ is the basic kernel size and $\beta$ is kernel rescale factor, a constant. We developed this mechanism to help RFB to deal with relations at different scale, and we observe that it produces significantly better result than vanilla RBFs.

### A.2  HYPER-PARAMETER SELECTION

We perform grid search for all methods based on their performance on the validation sets.

**Location Modeling**  The hyper-parameters of $theory$ models are based on grid search with $d^{(v)} = (32, 64, 128, 256)$, $d^{(x)} = (32, 64, 128, 256)$, $S = (4, 8, 16, 32, 64, 128)$, and $\lambda_{min} = (1, 5, 10, 50, 100, 200, 500, 1k)$ while $\lambda_{max} = 40k$ is decided based on the total size of the study area. We find out the best performances of different grid cell based models are obtained when $d^{(v)} = 64$, $d^{(x)} = 64$, $S = 64$, and $\lambda_{min} = 50$. In terms of $tile$, the hyper-parameters are selected from $c = (10, 50, 100, 200, 500, 1000)$ while $c = 500$ gives us the best performance. As for $rbf$, we do grid search on the hyper-parameters: $M = (10, 50, 100, 200, 400, 800)$ and $\sigma = (10^2, 10^3, 10^4, 10^5, 10^6, 10^7)$. The best performance of $rbf$ is obtain when $M = 200$ and $\sigma = 10^3$. As for $wrap$, grid search is performed on: $h = (1, 2, 3, 4)$ and $o = (64, 128, 256, 512)$ while $h = 3$ and $o = 512$ gives us the best result. All models use FFNs in their $Enc^{(x)}()$ except $wrap$. The number of layers $f$ and the number of hidden state neurons $u$ of the FFN are selected from $f = (1, 2, 3)$ and $u = (128, 256, 512)$. We find out $f = 1$ and $u = 512$ give the best performance for $direct$, $tile$, $rbf$, and $theory$. So we use them for every model for a fair comparison.

**Spatial Context Modeling**  Grid search is used for hyperparameter tuning and the best performance of different grid cell models is obtain when $d^{(v)} = 64$, $d^{(x)} = 64$, $S = 64$, and $\lambda_{min} = 10$. We set $\lambda_{max} = 10k$ based on the maximum displacement between context points and center points to make the location encoding unique. As for multiple baseline models, grid search is used again to obtain the best model. The best model hyperparameters are shown in () besides the model names in Table 3. Note that both $rbf$ and $scaled\_rbf$ achieve the best performance with $M = 100$.

### A.3  FIRING PATTERN FOR THE NEURONS

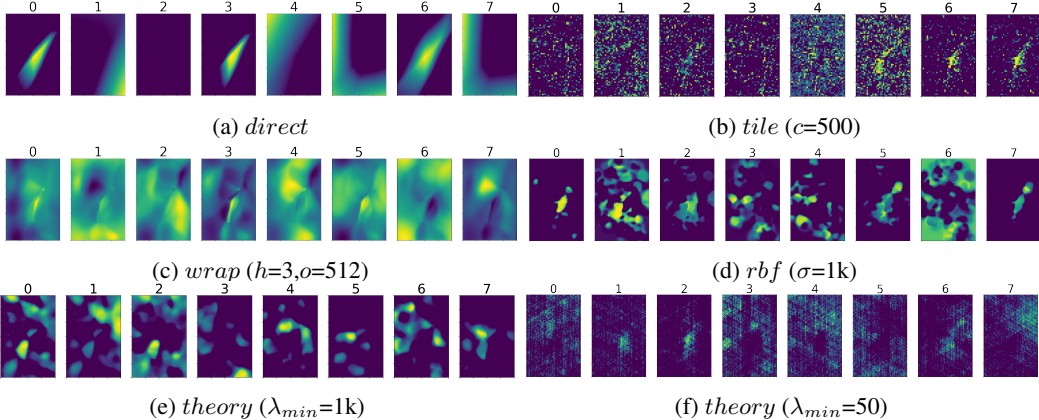

Figure 4: The firing pattern for the first 8 neurons (out of 64) given different encoders in location modeling.

### A.4 EMBEDDING CLUSTERING OF RBF AND THEORY MODELS

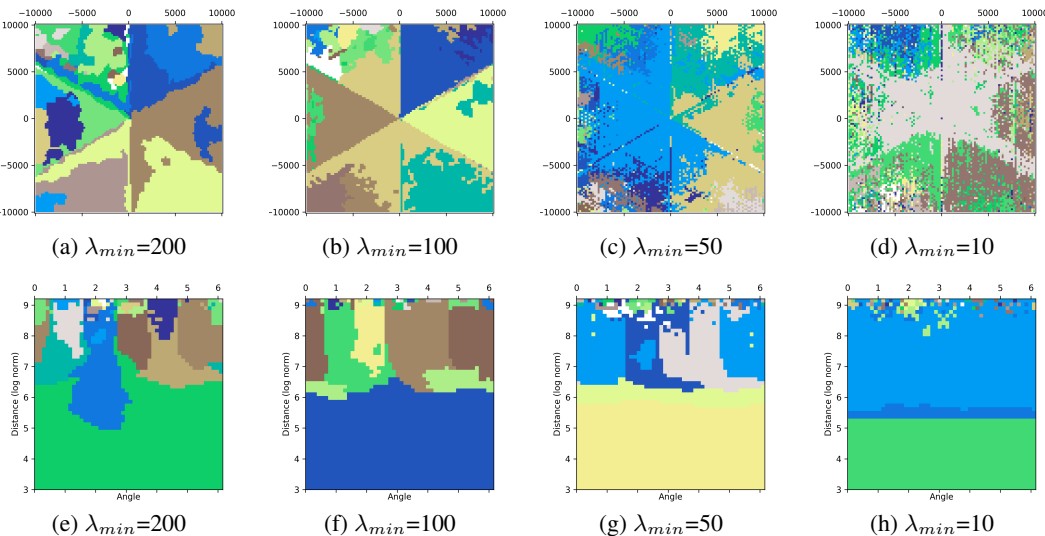

(a) $\lambda_{min}$=200    (b) $\lambda_{min}$=100    (c) $\lambda_{min}$=50    (d) $\lambda_{min}$=10

(e) $\lambda_{min}$=200    (f) $\lambda_{min}$=100    (g) $\lambda_{min}$=50    (h) $\lambda_{min}$=10

Figure 5: Embedding clustering in the original space of (a)(b)(c)(d) *theory* with different $\lambda_{min}$, but the same $\lambda_{max} = 10k$ and $S = 64$. (e)(f)(g)(h) are the embedding clustering results of the same models in the polar-distance space. All models use 1 hidden ReLU layers of 512 neurons.

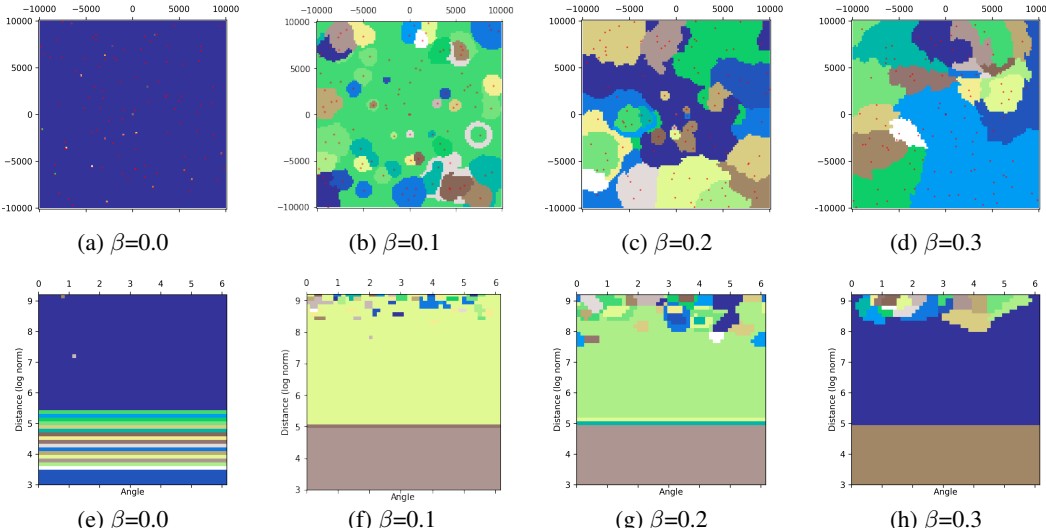

(a) $\beta$=0.0    (b) $\beta$=0.1    (c) $\beta$=0.2    (d) $\beta$=0.3

(e) $\beta$=0.0    (f) $\beta$=0.1    (g) $\beta$=0.2    (h) $\beta$=0.3

Figure 6: Embedding clustering of RBF models with different kernel rescalar factor $\beta$ (a)(b)(c)(d) in the original space; (e)(f)(g)(h) in the polar-distance space. Here $\beta$=0.0 indicates the original RBF model. All models use $\sigma$=10m as the basic kernel size and 1 hidden ReLU layers of 512 neurons.

