# OpenReview forum: "Multi-Scale Representation Learning  for Spatial Feature Distributions using Grid Cells"
_ICLR.cc/2020/Conference — Accept (Spotlight)_

### Official Review · AnonReviewer3 · 2019-10-22
**Official Blind Review #3**

**Rating:** 6

**Review:**

This paper presents a new method called "Space2Vec" to compute spatial embeddings of a pixel in a spatial data. The primary motivation of Space2Vec is to integrate representations of different spatial scales which could potentially make the spatial representations more informative and meaningful as features. Space2Vec is trained as a part of an encoder-decoder framework, where Space2Vec encodes the spatial features of all the points that are fed as input to the framework.

They conducted experiments on real world geographic data where they predict types of point of interests (POIs) at given positions based on their 1) locations (location-modeling) and 2) spatial neighborhood (spatial context modeling). They evaluated Space2Vec against other ML approaches for encoding spatial information including RBF kernels, multi-layer feed forward nets, and tile embedding approaches. Their results indicate that that Space2Vec approach performs  better (albeit marginally) than other ML methods.

I am giving this paper a weak reject rating mainly because of weak results and lack of motivation for location modeling problem (where their approach performs significantly better than baselines).   I explain my concerns below under detailed comments.

Detailed Comments:
1) Motivation of location modeling problem does not sound compelling enough to me, especially in the context of Point of Interest(POI) classification approach. I could not imagine any scenario where access to information from spatial neighborhood will be denied. If authors could present strong motivating examples for this problem and demonstrate the utility of their proposed approach in that setting, that will make the paper much stronger.
2) In spatial context modeling problem, the improvements in the results (Table 2) appear to be marginal(0.185 against 0.181, 25.7 against 25.3). Authors should try out more datasets to convincingly justify the superiority of their approach over other methods.

EDIT: AFTER RECEIVING AUTHOR'S RESPONSE
I am satisfied with author's response to my comments. I am updating my rating to Weak Accept.


**Experience Assessment:**

I do not know much about this area.

**Review Assessment: Checking Correctness Of Derivations And Theory:**

I assessed the sensibility of the derivations and theory.

**Review Assessment: Checking Correctness Of Experiments:**

I carefully checked the experiments.

**Review Assessment: Thoroughness In Paper Reading:**

I read the paper thoroughly.

---

> ### Author Response · Authors · 2019-11-12
> **Initial response to R3**
>
> We thank the reviewer for these insightful comments and suggestions. Our response to these comments are provided below:
>
> A. More tasks
> To demonstrate the generalizability of our space encoder and show how we will use it in other tasks, we utilized the proposed point space encoder model in a well-known computer vision task: fine-grained image classification. See details in our response to Reviewer 1 and added section of Appendix A.6. Briefly, we utilized the setup of Mac Aodha et al. [1] and replaced their location encoder with Space2Vec location encoder. The resulting model outperforms previous models as well as that of Mac Aodha et al. [1] on two datasets with significant sizes. This result clearly shows that Space2Vec is suitable for modeling the spatial prior information in fine-grained image classification problem which capture the species spatial distribution information.
>
> B. Motivation of the Location Modeling Problem:
> Density estimation is a long-standing problem in machine learning and statistics (Hastie et al. 2005). The task setup is inductive learning as no context examples are available after the training stage.  Popular approaches have evolved over time including discretization, kernel methods, and more recently neural net approaches. This work aims to propose a multi-scale grid cell encoding neural net, which outperforms the previous popular approaches.
>
> The location modeling problem represents the joint modeling the distribution of multiple classes in one model. It is not only technically interesting but is instrumental in many scientific problems. For example, a large portion of biodiversity data comes from endemic species. As the study of Myers et al. (2000) revealed, “as many as 44% of all species of vascular plants and 35% of all species in four vertebrate groups are confined to 25 hotspots comprising only 1.4% of the land surface of the Earth”. Building an inductive spatially explicit machine learning model for species distribution modeling helps us to predict what are other potential habitats outside known hotspots and plan for research and reservation efforts (Phillips et al, 2006; Zuo et al. 2008).
>
> C. Weak Result
> Our result is strong when only location information is used as model input (Table 1). As discussed in Gao et al [4], it is when there is a lack of visual clues that the grid cells of animals are the most helpful for their navigation. Location encoding is also our focus and core contribution.
>
> We agree with the reviewer that Space2Vec does not outperform RBF in spatial context modeling. We have updated the paper emphasizing that “Space2vec achieved a comparable performance with RBF on this task”. However, we want to point out that the scaled_RBF should not be considered as a baseline, because it is a specialized model we designed to help investigating how Space2Vec captures the distance and direction information in the space context modeling problem.
>
>
>
> References:
> 1. Mac Aodha, O., Cole, E. and Perona, P., 2019. Presence-Only Geographical Priors for Fine-Grained Image Classification. arXiv preprint arXiv:1906.05272.
> 2. Chu, G., Potetz, B., Wang, W., Howard, A., Song, Y., Brucher, F., Leung, T. and Adam, H., 2019. Geo-Aware Networks for Fine Grained Recognition. arXiv preprint arXiv:1906.01737.
> 3. Szegedy, C., Vanhoucke, V., Ioffe, S., Shlens, J. and Wojna, Z., 2016. Rethinking the inception architecture for computer vision. In Proceedings of the IEEE conference on computer vision and pattern recognition (pp. 2818-2826).
> 4. Gao, R., Xie, J., Zhu, S.C. and Wu, Y.N., 2018. Learning grid cells as vector representation of self-position coupled with matrix representation of self-motion. In Proceedings of ICLR 2019.
> 5. T Hastie, R Tibshirani, J Friedman, J Franklin. The elements of statistical learning: data mining, inference and prediction. The Mathematical Intelligencer 27 (2), 83-85
> 6. Myers, N., et al., 2000. Biodiversity hotspots for conservation priorities. Nature, 403(6772), p.853.
> 7. SJ Phillips, RP Anderson, RE Schapire, Maximum entropy modeling of species geographic distributions.  Ecological modelling, 2006 - Elsevier
> 8. Zuo, W., Lao, N., Geng, Y. and Ma, K., 2008. GeoSVM: an efficient and effective tool to predict species' potential distributions. Journal of Plant Ecology, 1(2), pp.143-145.

---

> > ### Comment · AnonReviewer3 · 2019-11-15
> > **Score Updated to Weak Accept**
> >
> > Thank you for responding back to the comments. I am satisfied with most of author's responses to my comments as well as the comments of other reviewers.
> >
> > I will recommend authors to add a few sentences in the main draft explicitly highlighting the motivation of location-modeling problem, specifically justifying one of the statements they made "The task setup is inductive learning as no context examples are available after the training stage".
> >
> > Location modeling does not sound like a smart formulation for an application like POI classification. The results from baselines of spatial context modeling problem are anyways better than the best performing approach for location modeling formulation. Ideally, I would have loved to see the results (for loc modeling part) on datasets from applications where spatial context is not available or hard to get and location modeling is indeed the genuine choice.

---

> > > ### Author Response · Authors · 2019-11-15
> > > **Thanks for the feedback**
> > >
> > > We will make this point clear in the final version, and experiment with more inductive learning tasks in our future work.

---

### Official Review · AnonReviewer2 · 2019-10-23
**Official Blind Review #2**

**Rating:** 8

**Review:**

The paper introduces Space2Vec, a space representation learning model. The work is motivated by the biological grid cell’s multi-scale periodic representations and the success of representation learning of NLP. So, the key idea behind the model is two-fold. On one hand, utilize the position information and the context associated with the position. On the other hand, the authors build a multiscale point space encoder based on Theorem 1 (in the paper), which was previously proved by Gao et al. (2019).
The multi-scale point feature encoder is novel. The experimental results turn out that the whole model is good at predicting features using only location information but does not outperform the RBF kernel (on validation) in terms of using spatial context modeling.
One core selling point of this paper is dealing with location distributions with very different characteristics. This is very well motivated at the beginning. More analysis/statistics would help better understand how the model "theory" wins Table one. See comments on experiments.

I have some comments/questions about model architecture and also experimental results/analysis.

1. Regarding Contextual Embedding
-- The encoder of this paper is not doing much “contextual embedding”.  The encoder typically encodes features of each position independently thus lead to very local embedding.
-- The location decoder would reconstruct the same (or similar) type of point features given embeddings of locations of the same type. As the distributions of different types of locations are very different, an encoder capable to deal with multi-scale data is crucial here.
-- The Spatial context decoder, like a context-dependent language model, would reconstruct the current position features, given the neighboring information.
-- Overall, unlike many existing pre-training models in NLP with deep encoders, the full model of this paper is with very local encoder, while the decoder does the most work of “gathering contextual information".
As you claim your model to be "a general-purpose space representation model", can you describe/specify how you would use your model for other tasks? Will you take some intermediate output of decoders to be representations? Or will you fine-tune the whole encoder-decoder?

2. For experiments:
a.	I do not prefer saying your method outperforms RBF in the “spatial context modeling” task when you getting worse validation set performance. It is interesting that RBF is stronger in terms of validation.
b.	Is it possible for the authors to do some statistics on the different types of locations? For your first task “location modeling”, should we expect to see that your model does not have very bad performance on certain types of locations while other non-multi-scale approaches do? This is trying to provide better support to one of your core contributions.
c.    Again, to claim the model can be widely applied, try more tasks?


To clarify my "experience assessment", I mean I read many related representation learning papers rather than specific papers related to GIS data.
My actual rating for the paper is between weak reject to weak accept (but the system does not have intermediate choices). I would like to hear the author's feedback to further revise the rating.



**Experience Assessment:**

I have read many papers in this area.

**Review Assessment: Checking Correctness Of Derivations And Theory:**

I assessed the sensibility of the derivations and theory.

**Review Assessment: Checking Correctness Of Experiments:**

I assessed the sensibility of the experiments.

**Review Assessment: Thoroughness In Paper Reading:**

I read the paper at least twice and used my best judgement in assessing the paper.

---

> ### Author Response · Authors · 2019-11-12
> **Statistic and Analysis of Location Modeling Problem:**
>
> Statistic and Analysis of Location Modeling Problem:
> Thanks for your suggestion. We have performed some statistical analysis on the location modeling task. More specifically, we divide all 1191 POI types into 3 different groups based on their typical radius r (defined as the x-position, where a renormalized Ripley’s K curve reaches 3.0, See Figure 1d):
> 1. Concentrated (r <=100m): POI types with concentrated distribution patterns;
> 2. Middle (100m < r < 200m): POI types with less extreme scales;
> 3. Even (r >= 200m): POI types with even distribution patterns
>
> We compare the performance of direct, tile, wrap, rbf baseline models and our theory model on these three groups. Detailed explanations can be seen in Appendix A.5 in our updated paper on openreview (Table 3), but briefly:
> 1. The two neural net approaches (direct and wrap) have no scale related parameter and are not performing ideally across all scales, with direct performs worse because of its simple single layer network.
> 2. The two approaches with built-in scale parameter (tile and rbf) have to trade off the performance of different scales. Their best parameter settings lead to close performance to that of Space2Vec at the middle scale, while performing poorly in both concentrated and even group;
>
> These observations clearly show that all baselines can at most well handle distribution at one scale but show poor performances in other scales. In contrast, Space2Vec’s multi-scale representation can handle distributions at different scales.

---

> > ### Comment · AnonReviewer2 · 2019-11-15
> > **Score updated**
> >
> > Thanks for your clarification and extra analysis.
> >
> > I agree with most of your response.
> > I think your paper proposes an interesting solution for studying spatial data representations and encouraging future research.
> > I've updated that score.

---

> ### Author Response · Authors · 2019-11-12
> **Initial response to R2**
>
> We thank the reviewer for these insightful comments and suggestions. We have grouped our answers into several topics with detailed explanations:
>
> A. The generalizability of Space2Vec space representation model (more tasks):
> To clarify our claim about "a general-purpose space representation model", the generalizability is about the encoder part of the model, while the contextual decoder is specific  to the presented task. To apply the space2vec encoder to other tasks, we will just encode location information into location embeddings and use these location embeddings in downstream tasks.
>
> To demonstrate the generalizability of our space encoder and show how we will use it in other tasks, we utilized the proposed point space encoder model in a well-known computer vision task: fine-grained image classification. See details in our response to Reviewer 1 and added section of Appendix A.6. Briefly, we utilized the setup of Mac Aodha et al. [1] and replaced their location encoder with Space2Vec location encoder. The result model outperforms previous models as well as that of Mac Aodha et al. [1] on two datasets with significant sizes.
>
>
> B. Comparison between RBF and Space2Vec in Spatial Context Modeling:
> We agree with the reviewer that Space2Vec does not outperform RDF in spatial context modeling. We have updated the paper emphasizing that “Space2vec achieved a comparable performance with RBF on this task”. However, we want to point out that
> 1. Actually the gains are small for all baseline approaches also. The reason is that we expect location encoding to be less important when context information is accessible. Similarly as discussed in (Gao et al, 2019), it is when there is a lack of visual clues that the grid cells of animals are the most helpful for their navigation.
> 2. The scaled_RBF should not be considered as a baseline, because it is a specialized model we designed to help investigating how Space2Vec captures the distance and direction information in the space context modeling problem.
>
> C. Use a “very local” encoder while having a decoder to gather contextual information:
> Our answers to this question are two folds:
> 1. We deliberately design a “very local” encoder so that it can be applied to many different tasks such as the location modeling task, spatial context modeling task as well as the fine-grained image classification tasks as we showed above. If the encoder considers the spatial context information, its application will be more limited.
> 2. In the graph neural network literature, several teams have discussed the drawbacks when the encoder considers too much contextual information, e.g., stacking too much graph convolution layers. For example, Li et al. [5] show that repeatedly applying Laplacian smoothing (stacking many graph convolutional layers) may have a negative impact on the model, because the model considers too much long-range information (large context) which makes the nodes indistinguishable. In this work, we prefer a more localized position encoder and use the decoder to capture the context information.
>
> We think this comment is very valuable and, in the future, we want to explore a way to combine both local and context information in the point encoder.
>
> D. Other issues about Contextual Embedding:
> We need to point it out that the location decoder reconstructs point features not only based on embeddings of locations of the same type but also based on embeddings of locations of other types. The spatial context of a POI is the K-nearest POIs of any type.
>
> Reference:
> 1. Gao, R., Xie, J., Zhu, S.C. and Wu, Y.N.. Learning grid cells as vector representation of self-position coupled with matrix representation of self-motion. In Proceedings of ICLR 2019, 2019
> 2. Mac Aodha, O., Cole, E. and Perona, P., 2019. Presence-Only Geographical Priors for Fine-Grained Image Classification. arXiv preprint arXiv:1906.05272.
> 3. Veličković, P., Cucurull, G., Casanova, A., Romero, A., Lio, P. and Bengio, Y., 2017. Graph attention networks. arXiv preprint arXiv:1710.10903.
> 4. Kipf, T.N. and Welling, M., 2016. Semi-supervised classification with graph convolutional networks. arXiv preprint arXiv:1609.02907.
> 5. Li, Q., Han, Z. and Wu, X.M., 2018, April. Deeper insights into graph convolutional networks for semi-supervised learning. In  AAAI 2018.

---

### Official Review · AnonReviewer1 · 2019-10-31
**Official Blind Review #3**

**Rating:** 6

**Review:**

The paper presents a model that learns an embedding/representation for spatial points (POI's). There are two specific things the representations are trying to encode - location modeling and spatial context modeling and the model tries to do it in multi-scale manner to increase the information/granularity of the learnt representations.

The experiments are performed on Yelp Data challenge which has 21,830 POI's with 1191 POI types. In the location context experiments authors show that by going after a smaller grid size we can get much better results compared to other methods while other methods like tile, wrap and rbf have more parameters causing overfitting. Similarly, on spatial context modeling we see better results.

Overall, the problem of learning vector representations for spatial points is interesting and useful and this paper has valuable contributions on how to do it.

One thing I would like to have seen to strengthen the paper further is the application of these representations in other tasks like image classification or recommendation systems or retrieval. The paper currently misses that.

For ex - https://arxiv.org/abs/1505.03873 uses location information to improve image classification, similarly can we use the representation learned through this method instead of positional coordinates and show that it helps the final task.

**Experience Assessment:**

I have published in this field for several years.

**Review Assessment: Checking Correctness Of Derivations And Theory:**

I assessed the sensibility of the derivations and theory.

**Review Assessment: Checking Correctness Of Experiments:**

I assessed the sensibility of the experiments.

**Review Assessment: Thoroughness In Paper Reading:**

I read the paper at least twice and used my best judgement in assessing the paper.

---

> ### Author Response · Authors · 2019-11-12
> **Apply Space2Vec to the fine-grained image classification task**
>
> Thank you for your valuable suggestion about adding another application of the proposed space representation model. We followed your suggestion and applied our Space2Vec to the task of  fine-grained image classification.
>
> The core idea is to use our Space2Vec to capture the spatial prior information of species distribution. We follow the exact experiment setup as Mac Aodha et al. [1], which had a similar inductive learning set up as our main result: using a location encoder to encode geographic coordinates into location embeddings. Mac Aodha et al. combined the location encoder with a pretrained InceptionV3 network to do image classification. Tang et al. [2] leveraged a diverse set of metadata (e.g. hashtags, ACS data), but they discretized latitude and longitude to train embedding for each space block, which is comparable to our tile-based baselines in Table 1 and 2. In fact, Mac Aodha et al. [1] already included Tang et al. [2] as a baseline for fine-grained image classification task. We also included Tang et al. [2] in Table 3.
>
> We utilized the original code of Mac Aodha et al. [1] and replaced their location encoder with our Space2Vec model. We picked two fine-grained image classification datasets of significant sizes, BirdSnap† (49,829 images spanning 500 species of North American birds) and NABirds† (555 categories with a total of 48,562 images) as example datasets which have been used in Mac Aodha et al. [1]. Experiment results showed that our grid and theory model can outperform previous models as well as the model of Mac Aodha et al. [1] on both datasets. For experiment results and details, please refer to Appendix A.6 in our updated paper on openreview.
>
> We believe that this additional task demonstrates the generalizability of our space representation model.
>
> References:
> 1. Mac Aodha, O., Cole, E. and Perona, P., 2019. Presence-Only Geographical Priors for Fine-Grained Image Classification. arXiv preprint arXiv:1906.05272.
> 2. Tang, K., Paluri, M., Fei-Fei, L., Fergus, R. and Bourdev, L., 2015. Improving image classification with location context. In Proceedings of the IEEE international conference on computer vision (pp. 1008-1016). https://arxiv.org/abs/1505.03873

---

### Public Comment · ~Kaishun_Zhang1 · 2019-12-25
**code**

great paper!
I will appreciate if you make the code online available.
look forward to your reply~

---

> ### Author Response · Authors · 2019-12-26
> **We will make our implementation public with the final version of the paper**
>
> Thanks for the feedback.

---

### Decision · Program_Chairs · 2019-12-19

**Decision:**

Accept (Spotlight)

**Comment:**

This paper proposes to follow inspiration from NLP method that use position embeddings and adapt them to spatial analysis  that also makes use of both absolute and contextual information, and presents a representation learning approach called space2vec to capture absolute positions and spatial relationships of places. Experiments show promising results on real data compared to a number of existing approaches.
Reviewers recognize the promise of this approach and suggested a few additional experiments such as using this spatial encoding as part of other tasks such as image classification, as well as clarification and further explanations on many important points. Authors performed these experiments and incorporated the results in their revisions, further strengthening the submission. They also provided more analyses and explanations about the granularity of locality and motivation for their approach, which answered the main concerns of reviewers.
Overall, the revised paper is solid and we recommend acceptance.